# *LSS* rs2254524 Increases the Risk of Hypertension in Children and Adolescents with Obesity

**DOI:** 10.3390/genes14081618

**Published:** 2023-08-13

**Authors:** Giuseppina Rosaria Umano, Grazia Cirillo, Giulia Rondinelli, Gianmaria Sanchez, Pierluigi Marzuillo, Stefano Guarino, Anna Di Sessa, Alfonso Papparella, Emanuele Miraglia del Giudice

**Affiliations:** Department of the Woman, the Child, of General and Specialized Surgery, University of Campania “Luigi Vanvitelli”, 80138 Naples, Italy; grazia.cirillo@unicampania.it (G.C.); giuliarondinelli90@gmail.com (G.R.); gianmaria.sanch@gmail.com (G.S.); pierluigi.marzuillo@unicampania.it (P.M.); stefano.guarino@policliniconapoli.it (S.G.); anna.disessa@unicampania.it (A.D.S.); alfonos.papparella@unicampania.it (A.P.); emanuele.miragliadelgiudice@unicampania.it (E.M.d.G.)

**Keywords:** blood pressure, children and adolescents, LSS rs2254524

## Abstract

Childhood obesity and its related comorbidities have become major health issues over the last century. Among these comorbidities, cardiovascular diseases, especially hypertension, are the most significant. Recently, a polymorphism affecting the activity of lanosterol synthase has been associated with an increased risk of hypertension in adolescents. In this study, we aimed to investigate the effect of *LSS* rs2254524 polymorphism on blood pressure in children and adolescents with obesity. We enrolled 828 obese children aged 6–17 years. Subjects carrying the A allele showed higher rates of systolic and diastolic stage I hypertension and stage II hypertension. Carriers of the A allele showed a 2.4-fold (95% C.I. 1.5–4.7, *p* = 0.01) higher risk for stage II hypertension and a 1.9-fold higher risk for stage I hypertension (95% C.I. 1.4–2.6, *p* < 0.0001). The risk was independent of confounding factors. In conclusion, *LSS* rs2254524 worsens the cardiovascular health of children and adolescents with obesity, increasing their blood pressure.

## 1. Introduction

Obesity has become a major health issue over the last century, for both children and adults. Over the past five decades, the prevalence of childhood overweight and obesity has steadily increased. Recent estimates report that about 254 million children and adolescents will live with obesity in 2030 [1]. Similarly, the rate of severe obesity in the pediatric age group has increased. About 25% of children and adolescents with obesity in European countries are diagnosed with severe obesity [2]. Moreover, the COVID-19 pandemic has led to weight gain in the general population compared to the pre-pandemic period as a consequence of the changes in lifestyles and habits, including reduced physical activity, increased screen time, and increased snacking [3]. In parallel with the spread of obesity, the associated metabolic diseases, including hypertension (HTN), dyslipidemia, hyperinsulinemia, elevated transaminases, and non-alcoholic fatty liver disease (NAFLD), have also increased [4]. Metabolic syndrome in adults is defined as a group of risk factors such as abdominal obesity, dyslipidemia, glucose intolerance, and hypertension. To date, no unified definition exists to assess risks or outcomes for children and adolescents. Due to age-related differences and different developmental stages in children and adolescents, the new definition of the International Diabetes Federation (IDF, 2005) is divided according to age groups. Children under six years of age were excluded due to insufficient data for this age group. Biochemical and anthropometric variables change with age and pubertal development. Consequently, in the absence of definitive data, the criteria adhere to the absolute values in the definition of adult IDF, except that waist circumference percentiles are recommended, and one cutoff is used for low–high-density lipoprotein cholesterol (HDL cholesterol). For children aged over 16 years, the IDF adult criteria can be used. In children younger than 10 years of age, metabolic syndrome should not be diagnosed, but it is important to emphasize the importance of weight loss to reduce health risks in the short and long terms [4].

Moreover, childhood obesity tracks into adulthood, increasing the risk of obesity and its related comorbidities in the long term, especially for children with severe obesity [5]. Epidemiological studies have shown a significant correlation between obesity and the risk of hypertension. According to the new guidelines of the American Academy of Pediatrics, in children and adolescents, the prevalence of high blood pressure is 15% [6]. In particular, as reported elsewhere, the prevalence of childhood hypertension ranges from 3.8% to 24.8% in young people with overweight and obesity. Hypertension in children with obesity may be accompanied by additional cardiometabolic risk factors, such as dyslipidemia and disordered glucose metabolism, which may have their own effects on blood pressure or represent comorbid conditions arising from the same unhealthy lifestyle behaviors. The grade of hypertension has a direct correlation with increased adiposity and a high BMI, since childhood obesity may lead to increased blood pressure in adulthood [6]. Notably, this risk appears to increase with the severity of obesity; there is a fourfold increase in blood pressure among those with severe obesity (BMI > 99th percentile) compared with a twofold increase in those with obesity (BMI 95–98th percentile) relative to normal-weight children and adolescents [6]. Previous longitudinal studies thoroughly established the role of blood pressure during childhood and adolescence to predict adult hypertension. Children with elevated blood pressure are more likely to develop hypertension in adulthood, with an increased risk of long-term comorbidities such as cardiovascular disease and chronic kidney disease [7,8,9]. In a recent meta-analysis conducted in 2020, which included 11 studies, it was highlighted that elevated blood pressure in childhood (3–18 years) was significantly associated with hypertension in adulthood. It also appears that increases of one standard deviation in systolic blood pressure and diastolic blood pressure in childhood were associated with a 1.71-fold higher risk (95% confidence interval 1.50–1.95) and a 1.57-fold higher risk (95% confidence interval 1.37–1.81) of hypertension during adulthood, respectively [8]. A study published in 2012 highlighted that childhood blood pressure levels, a patient’s overweight or obesity status, a history of parental hypertension, the parental occupational status, and 29 single-nucleotide polymorphisms in blood-pressure-associated genetic risk scores were independently related to hypertension at 21 to 27 years later in adulthood. Including this information has significantly improved the ability of the statistical model to predict adult hypertension. This suggests that a multifactorial approach could improve the identification of children with a high risk of adult hypertension [9]. The primary objective of nonpharmacologic and pharmacologic treatment of HTN in children and adolescents (where the therapy goal should be a reduction in systolic and diastolic blood pressure to <90th percentile and <130/80 mmHg in adolescents ≥13 years old) is achieving a blood pressure level that reduces the risk of target organ damage. In general, the lifestyle recommendations include, as is the case for adults, an active lifestyle and a diet rich in fruits, vegetables, low-fat dairy products, whole grains, fish, poultry, nuts, and lean red meat, a limited intake of sugar and sweets, and lower sodium intake [6]. In pediatric patients with obesity, considering the related risk factors, weight loss is part of the therapy. In addition to standard lifestyle approaches, weight-loss intensive care should involve regular contact with the patient and/or family and at least one hour of moderate to vigorous physical activity on a daily basis. If lifestyle changes do not sufficiently lower the patient’s blood pressure within about six months, drug treatment will be necessary. Oral therapy for persistent hypertension in children typically begins with an ACE inhibitor or calcium antagonist [6]. Both obesity and hypertension may share common causes, which are driven by a complex interaction of a patient’s genetic background and environmental factors.

A genome-wide association study (GWAS) analysis led to the identification of new loci associated with blood pressure, which are located not only in genomic regions known to be involved in blood pressure regulation but also in pathways and tissues that were previously thought to have no direct relationship with blood pressure. In obesity-related hypertension, different mechanisms interact in a complex network that also involves the sympathetic nervous system and the renin–angiotensin–aldosterone system and includes different pathological conditions, such as hyperinsulinemia and inflammation [10].

As demonstrated elsewhere, the evaluation of the correlation of the variants involved in hypertension in association with other complications related to hypertension, such as the mass and thickness of the left ventricle and coronary artery disease (CAD), has allowed researchers to define a genetic cardiovascular risk score. Furthermore, the identification of different genetic variants expressed in nephrons and glomeruli could link hypertension to renal physiology or kidney disease [11]. Moreover, associations between different blood pressure loci and the other complications that characterize metabolic syndrome (e.g., type II diabetes, obesity, and dyslipidemia) have been highlighted [12]. The understanding of the genetic scenario involved in blood pressure control has been further extended, with in silico analyses of genetic variants localized in coding regions and in transcriptional and post-transcriptional regulatory sites (the single-nucleotide variant and microRNA, miRNA) of several genes [13].

Although several GWAS have allowed for the identification of several genomic loci associated with systolic and diastolic blood pressure, there are still few reports identifying genetic variants that might indicate a causal relationship between childhood obesity and the risk of hypertension [14,15,16].

In a study carried out on Chinese children, the effect of obesity in association with gene variants related to blood pressure was analyzed. In particular, six single-nucleotide polymorphisms and the genetic risk score (GRS) were found to be significantly associated with higher systolic blood pressure or hypertension in obese children [14].

Recently, different research groups have focused their attention on the genetic and clinical differences between essential hypertension in non-obese and obese hypertensive children and adolescents, determining the prevalence of different allelic variants in single-nucleotide polymorphisms known to be involved in obesity and hypertension, although by a mechanism that has yet to be elucidated [15,16].

In a recent study carried out on adolescents, selected genetic variants were analyzed to evaluate their role in the regulation of blood pressure [17]. In particular, higher diastolic and systolic levels of blood pressure were associated with the missense variant of the lanosterol synthase gene (*LSS*), rs2254524 (Val642Leu; C > A), located in exon 20. This protein is a key enzyme in steroid biosynthesis, and it catalyzes the cyclization of (S)-2,3 oxidosqualene to lanosterol, a reaction that forms the sterol nucleus. The gene consists of 23 exons encoding a protein of 732 amino acids (83-kd). Thomas et al. found that it consists of two α/α barrel domains that are connected by loops and three rings of smaller β structures. The large cavity of the active site is at the center of the molecule between domains 1 and 2. The N-terminal region fills the space between the two domains and may work to stabilize their relative orientations [18]. A functional study in human adrenocortical cells showed that cells transfected with the Val642Leu mutant exhibited significantly higher enzymatic activity and endogenous ouabain (EO, a steroid hormone associated with hypertension) concentrations than cells expressing the wild-type protein [19]. More recently, a knock-in mouse model carrying the Lss V643L (LssV643L/V643L), homologous to human V642L, associated the role of the Lss gene in the regulation of blood pressure with an increased systolic blood pressure responsiveness to salt intake [20]. Based on the increasing evidence for a close association between obesity and hypertension since childhood, we aimed to examine the effect of the *LSS* rs2254524 polymorphism in children and adolescents with obesity and to investigate the potential interaction between this polymorphism and measures of adiposity.

## 2. Materials and Methods

Caucasian children and adolescents attending the obesity clinic of our Pediatric Department from January 2008 to January 2016 were retrospectively and consecutively enrolled. The study was conducted in compliance with the principles of the Declaration of Helsinki. The ethical committee of the University of Campania “Luigi Vanvitelli” approved the study (834/2016). Written informed consent was collected before any procedure was undertaken. The parents, legal guardians, and adolescents were asked to give their consent to participate in the study. Patients were eligible if they had a BMI ≥ 95th percentile for age and sex according to the reference charts [21], and if they were not taking medications affecting their blood pressure and weight status. All patients underwent a complete clinical and anthropometrical examination. Children and adolescents were weighed in their undergarments using a balance beam scale. Height was measured using a Harpenden stadiometer. BMI was calculated as weight (kg)/height^2^ (m^2^). The patients’ waist circumferences were measured at the midpoint between the lowest rib and the iliac crest, and the average of two values was obtained. The ratio between waist and height (WHR) in centimeters was calculated as an indirect measure of abdominal fat. Other measures of adiposity were calculated. The tri-ponderal mass index (TMI) was calculated as weight (kg)/height^3^ (m^3^) [22]. A height–weight formula proposed by Hudda et al. was calculated as previously described [23,24]. Blood pressure was measured three times, and the average value of the measurements was recorded. Blood pressure was defined according to the AAP 2017 guidelines as normal (blood pressure < 90th percentile in children aged <13 years and blood pressure <120/80 mmHg in adolescents aged ≥13 years); elevated (blood pressure ≥90th to <95th percentile in children aged <13 years and blood pressure ≥120/<80 mmHg to 129/<80 mmHg in adolescents aged ≥13 years); stage I hypertension (blood pressure ≥95th percentile in children aged <13 years and blood pressure ≥130/80 mmHg to 139/89 mmHg in adolescents aged ≥13 years); and stage II hypertension (blood pressure ≥95th percentile + 12 mmHg in children aged <13 years and blood pressure ≥140/90 mmHg) [25].

After fasting overnight, the children and adolescents underwent a blood sample test to measure their fasting plasma glucose and insulin. Immunoreactive insulin was assayed via IMX (Abbott Diagnostics, Santa Clara, CA, USA). The mean intra- and inter-assay coefficients of variations were 4.7% and 7.2%, respectively. The homeostasis model assessment for insulin resistance (HOMA-IR) was used as a measure of insulin resistance and was calculated as previously described [26].

Genomic DNA was extracted from peripheral whole blood using a DNA extraction kit (Promega, Madison WI, USA) according to the manufacturer’s instructions. Patients were genotyped for *LSS* L642V (rs2254524) using a Taqman allelic discrimination assay, which allows us to genotype the two possible variants at the single-nucleotide polymorphism site by measuring the change in the fluorescence of the dyes associated with the probes on an ABI 7900HT real-time PCR system (Thermo Fisher Scientific, Inc.; Waltham, MA, USA). Predesigned assay primers and probes (assay ID: C 3270849_40) were purchased from Thermo Fisher Scientific, Inc. Reactions were performed using the following condition: 2 min at 50 °C, 10 min at 95 °C, 15 secs at 95 °C, and 1 min at 60 °C for 40 cycles. The genotype of a subset of random samples was confirmed by PCR amplification and Sanger sequencing using the following primers: forward, 5′-GACCTTCCTTGGTGTGA-3′, and reverse 5′-GTCCCTCCTCTACCCAA-3′. PCRs were carried out using the following conditions: denaturation at 95 °C for 5 min followed by 35 cycles of 30 s at 94 °C, 30 s at 58 °C, and 30 s at 72 °C.

A Chi-square test was performed to test whether the genotypes were in Hardy–Weinberg equilibrium and for differences in categorical variables. Continuous variables were checked for normal distribution with the Kolmogorov–Smirnov test. Differences between genotypes for continuous variables were tested using the Kruskal–Wallis or ANOVA tests according to distribution. Univariate logistic regression was performed to assess the odds ratio for hypertension. Multivariate logistic regression was performed, including age, gender, and the BMI Z-score as covariates. The interaction between polymorphism and adiposity was assessed using a multivariate logistic regression analysis with the genotype (CC subjects as reference groups) and BMI Z-score, TMI, WHR, and height–weight formula. Data are expressed as the mean ± standard deviation (DS). All the analyses were performed using SAS^®^ on Demand for Academics (SAS Institute Inc., Cary, NC, USA).

## 3. Results

We observed 896 patients, of whom 68 were excluded because they did not meet the inclusion criteria. A total of 828 children and adolescents (51.2% males) with a mean age of 10.96 ± 2.62 and a mean BMI Z-score of 2.71 ± 0.61 were enrolled. The frequencies of the *LSS* rs2254524 polymorphism were in Hardy–Weinberg equilibrium (*p* > 0.05). In total, 82 patients were homozygous for the A allele (9.9%), 360 were heterozygous (44.2%), and 366 were homozygous for the C allele (45.9%). Table 1 reports the characteristics of the whole cohort. Elevated blood pressure was found in 25.6%, stage I hypertension in 29.5%, and stage II in 5.2% of subjects.

The three genotype groups did not differ in terms of age, BMI Z-score, sex, WHR, TMI, or height–weight formula (see Table 2). Subjects who were homozygous and heterozygous for the A allele showed significantly higher rates of stage I and stage II hypertension (*p* = 0.0001 and *p* = 0.006, respectively, see Table 2). Moreover, subjects who were homozygous for the C allele showed a significantly lower prevalence of both systolic (*p* < 0.0001) and diastolic (*p* = 0.049) stage I hypertension.

A univariate logistic regression analysis showed that subjects carrying the A allele had an odds ratio of 1.4 (95% C.I. 1.05–1.8, *p* = 0.02) of showing an elevated risk of stage II hypertension. Moreover, carriers of the A allele presented a 1.9-fold higher risk for stage I hypertension (95% C.I. 1.4-2.6, *p* < 0.0001), and a 2.4-fold higher risk for stage II hypertension (95% C.I. 1.2–4.7, *p* = 0.01). Finally, the risk for systolic stage I hypertension and diastolic stage I hypertension was increased in A allele carriers compared to subjects homozygous for the C allele (odds ratio 2.1, 95% C.I. 1.5–2.9, *p* < 0.0001; odds ratio 1.6, 95% C.I. 1.1–2.3, *p* = 0.02). These differences were independent from the effects of the confounding factors (see Table 3). In addition, the interaction between genotype and adiposity measures for the risk of hypertension was investigated. The genotype interacted with the BMI Z-score and WHR for the risk of stage I hypertension (*p* = 0.052 and *p* = 0.059, respectively) and diastolic stage I hypertension (*p* = 0.09 and *p* = 0.007, respectively). No other interactions were observed between genotype and TMI and the height–weight formula.

## 4. Discussion

Different neurohormonal systems regulate blood pressure to ensure the proper perfusion of tissues and organs. Multiple mechanisms underlie the association between obesity and hypertension, and they have not been completely defined yet. An unhealthy lifestyle, increased activity of the sympathetic nervous system, systemic low-grade inflammation, and increased renal sodium retention are involved in both conditions [27]. The body mass index (BMI) and body composition, especially visceral obesity, and the relationship between lean body mass (muscle) and the amount of adipose tissue are among the main determinants of blood pressure values in the population [28]. Pediatric patients with obesity display higher rates of elevated blood pressure, with long term consequences for adult health and increased mortality for cardiovascular disease [4].

The association between elevated blood pressure in young adulthood and cardiovascular disease later in life has been evaluated by several relevant studies. It was shown that elevated blood pressure in childhood or adolescence was significantly associated with intermediate markers of cardiovascular disease, such as a high PWV (pulse wave velocity), a high cIMT (carotid intima-media thickness), and LVH (left ventricular hypertrophy) in adulthood, and with all-cause mortality.

These results highlight the importance of having a blood pressure that falls within the range of normal values in children and adolescents [29].

Among the metabolic risks associated with obesity, hyperinsulinemia represents the most common underlying pathogenic mechanism. Hyperinsulinemia increases the sympathetic adrenergic drive, but, in healthy individuals, it is offset by a decrease in peripheral vascular resistance. Instead, in the state of chronic hyperinsulinemia that characterizes obesity, the peripheral vascular resistance may not decrease to the same extent. Different studies have shown that increased sympathetic activity, caused by chronic hyperinsulinemia and inflammation, promotes insulin resistance and increases peripheral vascular resistance. It is suggested that sympathetic activation is a key factor in the development of elevated blood pressure in obese patients [28]. However, the genetic background also plays an important role in the regulation of blood pressure levels.

In the present study, we demonstrated that *LSS* rs2254524 polymorphism increases the risk for hypertension in children and adolescents with obesity.

Among the participants in the study group, *LSS* rs2254524 polymorphism was strongly associated with an increased risk of exhibiting stage I and stage II hypertension. Similar results were observed by Bigazzi et al. in Italian adolescents. In this study, *LSS* rs2254524 polymorphism, alongside other genetic variants, was associated with elevated blood pressure. However, these data were based on previous cut-off values that have been substituted by the AAP 2017 guidelines.

Notably, in our study, the risk for hypertension was independent of the severity of obesity. However, we observed a significant interaction between the BMI Z-score and WHR, with the presence of at least one A allele in heightening the risk for stage I hypertension. Therefore, we might speculate that *LSS* rs2254524 polymorphism influences the levels of blood pressure and that its effect is enhanced by adiposity, especially visceral adiposity. Notably, in our cohort, AA subjects showed lower levels of insulin and insulin resistance. This finding also supports the hypothesis that polymorphism exerts a specific effect on blood pressure levels, independent of insulin resistance, and that it is mostly derived from renal sodium retention. Lanosterol synthase is a key enzyme in cholesterol biosynthesis; it catalyzes the cyclization of oxidosqualene in lanosterol. This pathway finally leads to endogenous ouabain (EO) synthesis. Endogenous ouabain regulates the Na+-K+ pump [30], limiting the dipping of the blood pressure in the event of salt depletion. A functional study showed that the Val642Leu mutant exhibited significantly higher enzymatic activity and EO than the wild-type protein [19]. Moreover, a study by Lanzani et al. reported that hypertensive subjects who were homozygous for the A allele displayed a more pronounced reduction in blood pressure after consuming a low-salt diet compared to AC and CC subjects [24]. Nevertheless, these findings have not been verified in pediatric cohorts. However, the evaluation of the *LSS* rs2254524 genotype might aid in the development of a personalized approach to hypertension, even in childhood, particularly for at-risk populations such as children and adolescents with obesity; it may also enable the optimization of the nutritional intervention in treatments.

The first-line treatment for hypertension in children and adolescents with obesity consists of lifestyle changes by promoting a healthy diet and physical activity [6]. In addition, a specific diet has been proposed in the past to specifically address hypertension. The Dietary Approaches to Stop Hypertension (DASH) has been widely studied in adult cohorts for the management of hypertension [31]. The DASH eating plan includes foods with antihypertensive properties. In particular, this diet improves the intake of vegetables, fruits, whole grains, nuts, lean proteins, and legumes, and reduces the intake of high-fat foods, added sugar, and salt [31]. Adherence to the DASH eating plan can be assessed using the DASH score proposed by Fung et al. [32]. The DASH score attributes points for a high intake of healthy foods (namely, vegetables, legumes, fruit, and whole grains) and a low intake of red processed meats, salt, and sweetened beverages. Each component score is summed up to calculate the overall DASH score, with high scores indicating a good adherence to the DASH plan [32]. A recent meta-analysis investigated the effectiveness of adherence to the DASH diet in reducing the risk of hypertension in adult cohorts. The authors identified 12 trials and observed that high adherence to the DASH plan was associated with a lower odds ratio and hazard ratio of hypertension compared to the low-adherence group [33]. Scientific evidence for pediatric age is limited. However, it has been reported that the DASH score negatively correlates with blood pressure in children aged five to seven years [34]. In this study, for each 10-unit increase in the DASH score, a 0.7 mmHg reduction in systolic blood pressure was registered. The effect of diet was independent of weight loss. Moreover, the authors found an interaction between a genetic risk score for hypertension and the DASH score [34]. These findings suggest a gene–diet interaction that might be applied in clinical practice, as children with a high genetic risk might derive greater benefits from a specific diet.

This study has certain limitations. The cross-sectional design limits the possibility of evaluating the long-term effects of genotype on blood pressure. In addition, given the complexity of the regulation of blood pressure, other genetic polymorphisms might affect this trait. However, we have not included other genetic single-nucleotide polymorphisms in our analysis; therefore, other uninvestigated genetic influences should be hypothesized. Moreover, different nutritional interventions should be investigated to better clarify the effect of genotype on a low-salt diet compared to standard nutritional interventions for children with obesity.

In conclusion, these findings support the role of *LSS* rs2254524 polymorphism in worsening the cardiovascular comorbidities seen in pediatric obesity. More studies that investigate a more personalized patient-centered approach are recommended for pediatric age groups in particular.

## Figures and Tables

**Table 1 genes-14-01618-t001:** Characteristics of the study population.

Parameter	
Age	11.14 (2.44)
Sex (M, %)	51.1
Genotype (CC, AC, AA, %)	45.9, 44.2, 9.9
BMI Z-score	2.68 ± 0.57
WHR	0.61 ± 0.060
TMI	21.29 ± 2.77
Height–weight formula	31.64 ± 4.59
Systolic blood pressure-SDS	0.88 ± 1.08
Diastolic blood pressure-SDS	0.46 ± 0.75
Elevated blood pressure (%)	25.6
Stage I hypertension (%)	29.5
Stage II hypertension (%)	5.2

Legend: data are expressed as the mean ± standard deviations and frequencies. Abbreviations: TMI: tri-ponderal mass index; WHR: waist-to-height ratio.

**Table 2 genes-14-01618-t002:** Clinical, anthropometrical, and biochemical characteristics according to genotype.

Parameter	AA (*n* = 82)	AC (*n* = 366)	CC (*n* = 380)	*p*-Value
Age	10.87 ± 2.40	11.11 ± 2.47	11.22 ± 2.43	0.34
Sex (M, %)	53.7	50.3	51.2	0.85
BMI Z-score	2.69 ± 0.48	2.71 ± 0.57	0.62 ± 0.57	0.35
WHR	0.62 ± 0.05	0.61 ± 0.06	0.62 ± 0.06	0.75
TMI	21.48 ± 2.40	21.26 ± 0.57	21.29 ± 2.73	0.89
Height–weight formula	31.64 ± 2.40	31.63 ± 4.56	31.68 ± 4.51	0.79
Fasting glucose (mg/dL)	80.87 ± 7.76	80.63 ± 8.78	80.92 ± 8.25	0.23
Fasting insulin (mcUI/dL)	21.16 ± 15.16	25.91 ± 22.72	25.89 ± 19.15	**0.02**
HOMA-IR	4.22 ± 3.12	5.18 ± 5.02	5.14 ± 3.91	**0.04**
Elevated blood pressure (%)	34.2	24.8	24.4	0.17
SBP stage I hypertension (%)	15.9	18.8	15.7	**<0.0001**
DBP stage I hypertension (%)	15.9	18.8	12.6	**0.049**
Stage I hypertension (%)	32.9	36.8	22.7	**0.0001**
Stage II hypertension (%)	3.7	8.0	2.9	**0.006**

Legend: statistically significant p-values are reported in bold. Data are expressed as the mean ± standard deviations and frequencies. Abbreviations: DBP: diastolic blood pressure; HOMA-IR: homeostasis model assessment for insulin resistance; SBP: systolic blood pressure; TMI: tri-ponderal mass index; WHR: waist-to-height ratio.

**Table 3 genes-14-01618-t003:** Logistic regression analysis for hypertension according to genotype.

Outcome	OR	95% C.I.	*p*	Adjusted *p*
Elevated blood pressure	1.2	0.8–1.5	0.49	-
Systolic stage I hypertension (%)	2.1	1.5–2.9	**<0.0001**	**<0.0001**
Diastolic stage I hypertension (%)	1.6	1.1–2.3	**0.02**	**0.03**
Stage I hypertension (%)	1.9	1.4–2.6	**<0.0001**	**<0.0001**
Stage II hypertension (%)	2.4	1.2–4.7	**0.01**	**0.01**

The multivariate analysis included the effects of age, sex, and the BMI Z-score.

## Data Availability

Data are available upon request to the corresponding author.

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
