# Peer review of "LSS rs2254524 Increases the Risk of Hypertension in Children and Adolescents with Obesity"

_genes, 2023, doi:10.3390/genes14081618_

Round 1
Reviewer 1 Report
Review for the paper “LSS rs2254524 increases the risk of hypertension in children 2 and adolescents with obesity”
The authors studied the effect of LSS rs2254524 (Val642Leu) polymorphism on blood pressure in children and adolescents with obesity.
They found that this polymorphism,increasing the levels of blood pressure and that represent increased risk factor for stage I and II hypertension.
Comments
1. The research design is appropriate.
2. The methods were well described but in order to be reproduce by others laboratories, please mention the conditions for genotyping using a Taqman allelic discrimination assay.
3. Please mention the provenience of the ABI 7900HT Real Time PCR system.
4. The results are clearly presented.
5. Did you investigated other polymorphisms located in genes from cholesterol metabolism? If not, maybe this is another limitation of the study.
6. Please provide the number of the ethical committee approval.
Author Response
- The research design is appropriate.
Answer: thank you for your comment.
- The methods were well described but in order to be reproduce by others laboratories, please mention the conditions for genotyping using a Taqman allelic discrimination assay.
Answer: thank you for your suggestion, we have specified it in the methods, please see page 3 lines 130-132.
- Please mention the provenience of the ABI 7900HT Real Time PCR system.
Answer: thank you for your suggestion, we have specified the provenience in the methods section, please see page 3 line 130.
- The results are clearly presented.
Answer: thank you for your comment.
- Did you investigated other polymorphisms located in genes from cholesterol metabolism? If not, maybe this is another limitation of the study.
Answer: thank you for your comment. We have included only LSS polymorphism in our analysis. According to your comment, we have added this limitation in the discussion section, please see page 6 lines 242-246.
- Please provide the number of the ethical committee approval.
Answer: thank you for your comment, we have added the number in the methods section, please see page 3 line 98.

Reviewer 2 Report
In the manuscript (MS) “LSS rs2254524 increases the risk of hypertension in children and adolescents with obesity” Giuseppina Rosaria Umano and colleagues present a single centre observational study in obese children and adolescents that evaluated association of lanosterol synthase gene (LSS) gene polymorphism with hypertension and show that the polymorphism is associated with higher blood pressure independently of obesity. The potential involvement of LSS in increased synthesis of endogenous oubain as a pathophysiological link between the polymorphism and elevated blood pressure in this cohort of young patients is especially interesting and should be further explored in the future studies.
The MS is generally well written and concise, with references to relevant and current literature. In my view the results are interesting and clinically relevant. My main concern is related to description of patients’ recruitment and collection of informed consent, as outlined below.
Major points:
1. Please, confirm that the study was carried out along the Declaration of Helsinki and that informed consent was provided by parents/legal guardians (and depending on the age of participants also by adolescents involved in the study).
2. The ethical approval number should be provided.
3. Please, confirm if participants were recruited prospectively or retrospectively.
4. When were patients recruited into the study?
5. How many patients were approached, but did not agree to take part in the study or had to be excluded due to other causes?
Minor points:
1. Table 2 and Table 3 – rows are skewed and specific values do not align with presented parameters.
Author Response
Major points
- Please, confirm that the study was carried out along the Declaration of Helsinki and that informed consent was provided by parents/legal guardians (and depending on the age of participants also by adolescents involved in the study).
Answer: thank you for your comment, we have specified it in the methods section. Please see page 2 and 3 lines 96-99.
- The ethical approval number should be provided.
Answer: thank you for your suggestion, we have added the protocol number in the methods section, please see page 3 line 98.
- Please, confirm if participants were recruited prospectively or retrospectively.
Answer: thank you for your comment, we have retrospectively recruited our patients. We have added this information in the methods section, please see page 2 line 95.
- When were patients recruited into the study?
Answer: thank you for your comment, we recruited our patients from January 2008 to January 2016. We have added this information in the methods section, please see page 2 line 95.
- How many patients were approached, but did not agree to take part in the study or had to be excluded due to other causes?
Answer: thank you for your suggestion, we have approached 896 patients, of whom 68 were excluded because did not meet the inclusion criteria (use of medications, secondary forms of obesity). We have added this information in the results section, please see page 3 line 148-149.
Minor points:
- Table 2 and Table 3 – rows are skewed and specific values do not align with presented parameters.
Answer: thank you for your suggestion, we have modified the tables.
